# Realising the full potential of data-enabled trials in the UK: a call for action

Matthew R Sydes [1] , Yolanda Barbachano, [2] Louise Bowman [3] , Tom Denwood [4] , Andrew Farmer [5] , Steph Garfield-Birkbeck [6] , Martin Gibson [7] , Martin C Gulliford [8,9] David A Harrison [10] , Catherine Hewitt [11] , Jennifer Logue [12] , Will Navaie, [13] John Norrie [14] , Martin O'Kane, [2] Jennifer K Quint [15] , Jo Rycroft-Malone [12,16] , Jonathan Sheffield, [17] Liam Smeeth [18] , Frank Sullivan [19,20] Juliet Tizzard, [13] Paula Walker, [2] John Wilding [21] , Paula R Williamson [22] , Martin Landray [23,24,25] Andrew Morris [26] Rhoswyn R Walker [26] Hywel C Williams [27,28] , Janet Valentine, [29] The Data Enabled Trials Group Workshop Group members

ML, AM, RRW, HCW and JV are joint senior authors.

**Correspondence to**
Professor Matthew R Sydes; m.sydes@ucl.ac.uk

## ABSTRACT

**Rationale** Clinical trials are the gold standard for testing interventions. COVID-19 has further raised their public profile and emphasised the need to deliver better, faster, more efficient trials for patient benefit. Considerable overlap exists between data required for trials and data already collected routinely in electronic healthcare records (EHRs). Opportunities exist to use these in innovative ways to decrease duplication of effort and speed trial recruitment, conduct and follow-up.

**Approach** The National Institute of Health Research (NIHR), Health Data Research UK and Clinical Practice Research Datalink co-organised a national workshop to accelerate the agenda for 'data-enabled clinical trials'. Showcasing successful examples and imagining future possibilities, the plenary talks, panel discussions, group discussions and case studies covered: design/feasibility; recruitment; conduct/follow-up; collecting benefits/harms; and analysis/interpretation.

**Reflection** Some notable studies have successfully accessed and used EHR to identify potential recruits, support randomised trials, deliver interventions and supplement/replace trial-specific follow-up. Some outcome measures are already reliably collected; others, like safety, need detailed work to meet regulatory reporting requirements. There is a clear need for system interoperability and a 'route map' to identify and access the necessary datasets. Researchers running regulatory-facing trials must carefully consider how data quality and integrity would be assessed. An experience-sharing forum could stimulate wider adoption of EHR-based methods in trial design and execution.

**Discussion** EHR offer opportunities to better plan clinical trials, assess patients and capture data more efficiently, reducing research waste and increasing focus on each trial's specific challenges. The short-term emphasis should be on facilitating patient recruitment and for postmarketing authorisation trials where research-relevant outcome measures are readily collectable. Sharing of case studies is encouraged. The workshop directly informed NIHR's funding call for ambitious data-enabled trials at scale.

There is the opportunity for the UK to build upon existing data science capabilities to identify, recruit and monitor patients in trials at scale.

## INTRODUCTION

Clinical trials will continue to be the gold standard for testing pharmacological, behavioural and policy interventions for the foreseeable future. The advent of the COVID-19 pandemic has raised the profile and importance of trials with the British public.[1] For a large majority of conditions, late-phase randomised controlled trials are commonly long-term endeavours that are increasingly expensive, particularly industry-led trials. There is considerable overlap between the data that triallists need to assess impact of interventions on patients and what is already routinely collected in patient's health records. These routinely routinely collected data are reflected in local, regional or national datasets, collectively labelled here as 'electronic health records' (EHRs) (box 1). It is equally self-evident that safely making these routinely collected data available to researchers for purposeful access could decrease duplication of effort, reduce costs and may speed answers from some trials. The value of EHR in clinical trials is increasingly recognised by many organisations, including the UK (UK) Government as highlighted in the Life Sciences Industrial Strategy.[2,3]

The National Institute for Health Research (NIHR), Health Data Research UK (HDR UK) and the Clinical Practice Research Datalink (CPRD) (box 2) brought together UK researchers for a 1-day workshop in

**Box 1    What are routinely collected electronic health records?**

There are vast amounts of data routinely being collected on patients, for example, in patient electronic health records (EHRs), as well as disease and patient registries. Such data are commonly called real-world data, reflecting that it is routinely collected while patients go about their regular lives, as opposed to being specifically collected in a clinical trial. Because the National Health Service (NHS) is a universal healthcare provider, free at point of care, the NHS EHR offers the possibility to collect the longitudinal medical history of 98% of the UK population who uses the NHS. EHR therefore provides an excellent research tool for studying population health. Primary care EHRs are recorded by general practitioners (GPs) and contain longitudinal demographic and clinical patient information encompassing symptoms, diagnoses, prescribing, laboratory tests and data from hospital visits and admissions. Primary care EHR can be collected and made available for research in near real time. Secondary care records capture interactions that take place in a specific hospital and are event based, not longitudinal. Secondary care records are coded for reimbursement purposes and not usually by clinicians. There is a time lag between patient hospital visits and secondary care EHR data being made available for research and National Health Service planning purposes.

**Box 2    National Institute for Health Research (NIHR), Health Data Research UK (HDR UK) and Clinical Practice Research Datalink (CPRD)**

**National Institute for Health Research**
NIHR, a virtual organisation, is the research arm of the National Health Service. It comprises an academy of trainees and senior investigators, and systems to make research processing easier and faster, including infrastructure, such as the clinical research network that plays a vital role in delivering research, and a number of research funding programmes. These include, for example, the NIHR Health Technology Assessment Programme, NIHR Health Services and Delivery Research and the NIHR Public Health Programme that fund prioritised health research for the public on behalf of the Department of Health and Social Care.

**Health Data Research UK**
HDR UK is the national institute for health data science. The institute was established in 2018 as a joint strategic investment (HDR UK is funded by the Medical Research Council, the health research departments of England, Scotland, Wales and Northern Ireland (NIHR, Chief Scientist Office, Health and Care Research Wales, HSC Research and Development, respectively), the Economic and Social Research Council, the Engineering and Physical Sciences Research Council, the British Heart Foundation and Wellcome.) with an aim to 'unite the UK's health data to make discoveries that improve people's lives'. Through initially six collaborative research sites across the UK and expanded to eight sites and seven specialist HDR Hubs, the Health Data Research Innovation Gateway and Health Data Research Alliance, the institute develops advanced health data science tools and technologies to analyse large-scale and diverse health data to solve complex health challenges. One of the priority themes is 'Better, faster, and more efficient clinical trials'.

**Clinical Practice Research Datalink**
CPRD is jointly sponsored by the Medicines and Healthcare products Regulatory Agency and NIHR and is part of the Department of Health and Social Care. CPRD is the UK Government's dedicated real-world research service, which collects deidentified patient data on a daily basis from a network of one in every five GP practices across the UK, encompassing 25% of the UK population. The CPRD database includes longitudinal primary care records on a total 60 million patients collected over CPRD's 30-year history, with a median patient follow-up time of 10 years. Approximately 40% of CPRD's GP practice network are research active.[16]

October 2018 to discuss how clinical trials might undergo a step-change in the UK through being 'data-enabled'. 'Data-enabled trials' were defined for the workshop as introducing efficiencies in the following ways: improved study planning and feasibility assessments; more efficient recruitment, facilitated by accessing EHR to identify potential participants; efficient assessment of treatment benefits and potential harms by directly capturing routine EHR into the trial dataset; and more efficient study analyses and interpretation, using standard EHR variables. Many lessons from the workshop are also relevant to clinical trials more broadly.

Informed by this workshop, NIHR commissioned a cross-programme call for 'Data-enabled trials'.[4] Table 1 contains a list of data-enabled trials that were funded as a result of this call.

While there is recognition that challenges exist to accessing and using all sources of EHR, great strides have already been made realising the potential for using available EHR in innovative ways to support the delivery of patient consented studies. The workshop's spirit was therefore to focus on what had already been achievable and to consider imaginable solutions to current challenges and further possibilities that could be developed in the future at scale. Speakers and facilitators were asked 'to demonstrate existing capability and opportunities in the UK for undertaking data-enabled clinical trials, in particular showcasing the use of routine data to support different components of trial design and delivery, and regulatory considerations'. The workshop, which was attended by researchers from around the UK, included plenary talks, panel discussions, group discussions and the presentation of selected case studies. The elements were broadly grouped as: design and feasibility; recruitment; conduct

and follow-up; collecting key data on benefits and harms; and analysis and interpretation. Here, we summarise and update some key issues that emerged in these areas in order to inform future discussions and research activities.

## FINDINGS
### Design and feasibility
Assessing the potential sufficiency of accrual is key to the planning and funding of all trials. Recruitment can be challenging both in prevalent and incident clinical conditions, particularly where potential participants are managing their chronic conditions in an environment away from active researchers. Robust and scalable processes to estimate the number of potential participants

**Table 1** Trials funded in summer 2020 by first round of NIHR data-enabled trials call

| NIHR panel | NIHR ref | Trial title | Chief investigator (co-chief investigator) | Trials unit | Planned start |
|---|---|---|---|---|---|
| HTA | NIHR130508 | UK-ROX: Evaluating the clinical and cost-effectiveness of a conservative approach to oxygen therapy for invasively ventilated adults in intensive care | Daniel Martin | The Intensive Care National Audit and Research Centre | Q3-2020 |
| HTA | NIHR130280 | DaRe2THINK: Preventing stroke, premature death and cognitive decline in a broader community of patients with atrial fibrillation using healthcare data for pragmatic research | Dipak Kotecha | Birmingham Clinical Trials Unit | Q4-2020 |
| HTA | NIHR130573 | VITALL Volatile vs Total intravenous Anaesthesia for major non-cardiac surgery – a pragmatic randomised controlled trial | Joyce Yeung | Warwick Clinical Trials Unit | Q4-2020 |
| HTA | NIHR130548 | ASYMPTOMATIC: Assessing Symptom-driven vs Maintenance Preventer Therapy for the Outpatient Management of Asthma In Children | Ian Sinha (Paula Willliamson) | Liverpool Clinical Trials Centre | Q1-2021 |
| HS&DR | NIHR130075 | In silico trials of targeted screening for abdominal aortic aneurysm using linked healthcare data: can the efficiency of the NHS Abdominal Aortic Aneurysm Screening Programme be improved whilst maintaining publicly acceptable levels of disease detection in an ethically acceptable manner | Matthew Bown | n/a | Q4-2021 |
| HS&DR | NIHR130581 | Cluster randomised trial to improve antibiotic prescribing in primary care: individualised knowledge support during consultation for general practitioners and patients | Tjeerd van Staa | Manchester Clinical Trials Unit | Q3-2020 |
| HS&DR | NIHR130107 | ATHENA-M: Observational study of age, test threshold and frequency on English national mammography screening outcomes | Sian Taylor-Phillips | n/a | Q4-2020 |

Eighteen applications were submitted to the NIHR data enabled trials call across three research committees. Four of nine applications to HTA accepted for funding; two of five applications to HS&DR accepted for funding with one additional application still to be decided; zero of three application to public health research accepted for funding.
HS&DR, Health Services and Delivery Research; HTA, Health Technology Assessment; NIHR, National Institute of Health Research.

that might be suitable for a particular study need to be established.

Historically, complex care pathways made it difficult to estimate and then locate potential trial participants, because they might enter the systems in a number of ways. Use of evidence from the EHR, where these are available, enables a more realistic estimate of the achievable sample size within specific geographical regions based on demographic and clinical information within EHRs. Key eligibility criteria such as demographic information, medication dose, laboratory results, timing of clinical events and comorbidities within the EHR allow rapid estimates of the number of potentially eligible people that can be flagged as suitable for a trial.

In the UK, general practitioners (GPs) are the gatekeepers to the National Health Service (NHS), and as such, primary care records hold information that includes demographic and clinical data derived from consultations within primary care as well as secondary care visits and from other sources such as laboratory tests. Primary care records, covering a patient's medical history, are therefore a rich resource to carry out feasibility assessments and to locate patients for studies that might take place in a range of settings. Timeliness of data is important. The primary care data that CPRD receives, encompassing 25% of UK population, is updated daily, and it forms the basis for feasibility assessments CPRD routinely carries out for academic and industry led trials.[5] To further advance the use of EHR in supporting clinical trials, the NHS DigiTrials Health Data Research Hub for Clinical Trials was established by HDR UK in 2019 to facilitate feasibility assessments in England, initially using secondary care Hospital Episode Statistics (HES).[6 7]

The usual method of locating patients for many studies involves manual searches of patient records at each GP practice. The COPE study,[8] a patient-consented longitudinal cohort study was one of the first studies in the UK exploring the use of EHR as a means of locating specific patients for enrolment into a study. The study, which sought to identify patients with chronic obstructive pulmonary disease (COPD) with prior exacerbation events (table 2) demonstrated that it is possible to use a primary care EHR database to screen, locate and recruit participants for research. This method 'provides access to a cohort of patients while minimising input needed by GPs and allows researchers to examine healthcare usage and disease burden in more detail and in real-life settings'.[8] CPRD carried out a centralised search for eligible patients in its database of pseudonymised patients, across a defined geographical region, and GPs in CPRD's network were supplied with a list of potentially eligible patients based on the study inclusion and exclusion criteria. The GP then carried out a clinical review and invited potentially suitable patients to attend specific centres in which recruitment and collection of bespoke data could be completed. This intermediate clinical validation step involving the GP confirming a patient's eligibility both facilitated locating high-quality patients and maintained patient confidentiality.

| Table 2 | Key points from COPE study |
|---|---|
| Study name (registration) | Characterisation of COPD Exacerbations using Environmental Exposure Modelling (COPE)[8] |
| Sponsorship | Academic – Imperial, MRC funded. |
| Clinical setting | Chronic obstructive pulmonary in primary care, UK only: national. |
| Design | Patient-consented longitudinal cohort study (non-randomised). |
| Size | N=130. |
| Key outcome measure | COPD exacerbations. |
| Rationale for presentation | Design and feasibility. |
| Key discussion points | The aim of the COPE study was to develop a method for predicting COPD exacerbations as a result of environmental exposures. Quality algorithmic query specification to identify patient cohorts in EHR relies on data quality. The algorithm for COPE was validated in a previous study by comparing the EHR records with additional information provided by GPs.[60] The validated query was applied centrally to search for patients in the CPRD database. GPs were then contacted and asked to take part in confirming patient eligibility from a prescreened patient list. Centralised searching of the EHR greatly minimised GP screening time in confirming patient eligibility. It was also possible for a GP to exclude patients who may not be suitable to take part (eg, due to caring for a sick relative). No algorithm could reliably account for these types of patient exclusions. Lessons from the study were that it is feasible to recruit patients from EHR and readily collect data for long-term follow-up, although this method does depend on GP participation. Using EHR meant far less effort than manual screening of GP records. Because of the specificity of the EHR and clinical validation steps, more appropriate patients were approached, which reduced time needed for patient screening. Patients approached were keen to be involved in the study and as a result the enrolment was higher than using conventional methods. |

COPD, chronic obstructive pulmonary disease; EHR, electronic health record; GPs, general practitioners.

A large proportion of trials fail to recruit to time and target,[9] and the funding wasted on these efforts may be better invested in modest pre-trial efforts to determine likely feasibility. Assessing potential numbers of patients can reasonably be done with near real time, longitudinal EHR (likely primary care datasets) or an administrative EHR record, like HES, which is cross-sectional and broad but has a time delay. Given the demonstrated value of using an EHR evidence-based approach to assessing feasibility, research funding bodies will need to give careful consideration to how such work-up activities can be supported as dedicated funding for research usually starts only after the work to assess feasibility has been attempted. Key points to emerge for study planning using EHR were the importance of using a validated algorithm to search the data and the benefits of incorporating a clinical review to further refine the specificity of the search. These approaches can result in a much higher proportion of suitable and engaged patients presenting for detailed eligibility assessment at study sites.

## Recruitment

Recruiting patients to research studies should be a standard part of routine practice. The majority of UK hospital trusts engage in research activity, and this is increasingly true in primary care with 40% of GP practices in England participating in research in 2018/2019.[10] However, for most practitioners, there is limited time to spend with patients and often not enough time to explain research so lower burden approaches must be found.[11 12] Searches of GP patient records for patients potentially suitable for clinical studies is not new. However, this is usually done by searching right through GP patient records on a local practice basis, which can be a burden for practice staff and introduce variability in interpretation of selection criteria. Centralised patient eligibility searches, such as those carried out by CPRD and North West eHealth, reduce the burden on practice staff by supplying the practice a list of potentially eligible patients located through standardised database searches against protocol inclusion and exclusion criteria.

Large-scale use of EHR for patient searches must take into account information governance regulations and the need to maintain patient confidentiality. One approach is to search pseudonymised patient data, where a patient's identity is unknown to the researchers, as is the case with CPRD recruitment services. Potentially suitable patients are reidentified by their GP, thereby maintaining confidentiality between the patient and the clinician. In some cases, specific regulatory approvals may be granted by to individual researchers to obtain access to identifiable EHR patient data in order directly invite them into a clinical trial. Alternative methods for recruiting patients into prospective trials include establishing a cohort of patients or volunteers who are willing to take part in trials, such as the NHS Scotland SHARE scheme[13] and COVID-19 vaccine trials portal[14]—volunteer datasets may contain phenotype or genotype data

not available in clinical or administrative datasets— and pre-consenting trial participants to be reapproached about other studies, such as those following on from the INTERVAL trial (NCT01610635) of blood donation timing.[15]

EHRs are a tool that can be used to speed the pairing of willing patients with suitable trials.[16] DECIDE is a trailblazing randomised pragmatic clinical effectiveness trial, involving a three-way partnership between academia, industry and CPRD, which recruited patients with type 2 diabetes in a routine primary care setting, randomised to the licenced intervention or standard of care.[17] The study involved centralised searches of CPRD's primary care data to enable GPs in the CPRD network across all four UK nations to readily enrol suitable patients into the study when they presented for their routine diabetes checks. The novelty of this pragmatic clinical effectiveness study was that the study clinical data were captured directly from the patient's primary care EHR and flowed into CPRD's interventional research services platform (IRSP). The trial including randomisation was remotely managed via this central digital platform. The EHR derived study data were supplemented by electronic patient-reported outcomes collected by patients remotely logging into the IRSP. Table 3 summarises the key lessons from DECIDE, using real-time EHR to facilitate recruitment and trial management. The DECIDE study demonstrates how it is possible to recruit patients across the UK into a randomised trial involving real-world type 2 diabetic patients, based on changing laboratory haemoglobin A1c (HbA1c) readings using dynamic data searches on primary care EHR data. This approach of using primary care data for time-bound patient recruitment offers great potential and generalisability for phase II to phase IV patient recruitment, independent of relying on busy clinicians identifying patients presenting at clinic visits.

The majority of clinical trial protocols involve inclusion or exclusion variables that are time dependent, for example, timeframe from diagnoses, procedures or prescriptions. Furthermore, many trials will have a limited window of opportunity during which recruitment can take place for any individual potential participant. If patients are identified too late, the research invitation could arrive after it stopped being relevant; anecdotally, such irrelevant invitations have been confusing and upsetting to vulnerable populations. In these circumstances, the EHR source used to located suitable patients for trial recruitment must contain patient information that is both current and dynamic.

Primary care data can provide a longitudinal and timely resource for locating patients suitable for both incident and prevalence trials. For example, CPRD's primary care records are updated daily and include hospital discharge information, so the system lends itself well to locating patients for trials taking place in primary or secondary care. GPs can also be alerted through system 'pop-ups' or dynamic daily refreshed patient lists, as for the DECIDE trial, which can support recruitment into incident trials

**Table 3** Key points from decide trial

| | |
|---|---|
| Trial name (registration) | DECIDE: Pragmatic Randomised 104 Week Multicentre Trial to Evaluate the Comparative Effectiveness of dapagliflozin and Standard of Care in Type-2 Diabetes[17] (NCT02616666) |
| Sponsorship | Industry then academic – AstraZeneca then Liverpool. |
| Clinical setting | Type 2 diabetes mellitus on metformin and needing better glucose control in primary care setting, UK only: national. |
| Design | Phase 4, randomised, open label, comparator trial. |
| Size | N>800 (ongoing). |
| Key outcome measure | Clinical success (glucose control) at 12 months. |
| Rationale for presentation | Recruitment. |
| Key discussion points | DECIDE is a clinical effectiveness trial comparing dapagliflozin, a sodium-glucose cotransporter-2 (SGLT2) inhibitor, to standard care (alternative medications such as sulfonylureas and DPP4 (Dipeptidyl-peptidase 4) inhibitors) in people with type 2 diabetes requiring second-line pharmacological treatment (after metformin). There was a limited window in which patients need to be approached before treatment is changed. |
| | CPRD managed the trial using their interventional research services platform that supported general practitioners (GPs) and patients logging into the system. Algorithms were developed to carry out daily searches of primary care EHR for potentially eligible patients meeting prespecified criteria. The primary care team then reviewed the search list and enrol eligible patients into the study at their next routine scheduled visit. |
| | Most of the data required for the trial was available in CPRD's routinely collected dataset, thereby facilitating collection of primary and secondary outcome measures as well as identification and recruitment. Patient-reported outcome measures were collected electronically. Trial monitoring could therefore be carried out centrally. |
| | Lessons from the trial are that EHR search algorithms are easy to set up, but feasibility assessments should take account of variability of coding and real-world clinical practice. Recruiting patients via EHR is achievable and allows easy collection of long-term data for follow-up, but as with all trials, it depends on GP and patient participation and engagement. Data enablement and platform technology allows real-time adaptation to real-time conditions. |

CPRD, Clinical Practice Research Datalink; EHRs, electronic healthcare records.

for patients who present at the GP practice with acute or unmanaged conditions.

HES data can also be used to locate patients who have visited hospital and may be suitable for a trial. The ORION-4 trial is using a combination of HES data searches and patient questionnaires to recruit 15 000 patients from across the UK with atherosclerotic cardiovascular disease.[18] However, HES data has its limitation as it may only be available some months (often 3 months) after a patient hospital visit, it does not include a patient's full medical history and may not include all potential patients, such where the condition was not severe enough to require a hospitalisation, for example, mild stroke. Some hospitals have developed their own EHR systems that support recruitment of patients presenting with acute conditions. However, there are currently no cross-hospital, interoperable electronic systems in use in secondary care that support dynamic recruitment.

### Conduct and follow-up

Access to routine EHR for individuals participating in a trial to monitor their important and relevant health outcomes has been a fundamental challenge for many trials that have attempted access late in the trial. This has often involved a lengthy application process via a changing approval system, frequently resulting in a considerable lag between requesting and accessing data. Many researchers have persisted, recognising the huge potential for efficient use of EHR in the conduct and follow-up of trials, particularly large-scale national trials.

ASCEND[19 20] was a UK-wide trial in 15 500 people with diabetes, evaluating the effect of aspirin and omega-3 fatty acids on the risk of cardiovascular events. The trial, whose key design features are summarised in table 4, used mail-based methods for recruitment, intervention delivery and follow-up, thereby avoiding the need for dedicated study sites. Initially, follow-up was by a paper-based questionnaire but, in the latter years of the trial, an online form was available for follow-up for those who preferred. Towards the end of the 7.5-year follow-up period, access to HES data was granted, which allowed missing outcome data to be recorded for participants who had not returned questionnaires. Preliminary unpublished comparisons in ASCEND between the questionnaire data and the EHR data showed good concordance, with analyses of the primary trial outcomes producing similar results irrespective of the data source used (L Bowman, personal communication, 2020).

Although the trial was beneficial for patients avoiding the need to travel to trial sites, it was labour intensive for the study team, involving mailing and phoning thousands

**Table 4** Key points from Salford Lung Study

| | |
|---|---|
| Trial name (registration) | Salford lung Study[21] (Extension=NCT03152669) |
| Sponsorship | Industry – GlaxoSmithKline. |
| Clinical setting | Chronic obstructive pulmonary, UK only: regional. |
| Design | Early-phase, cluster randomised trial run in one geographic location. |
| Size | N=7200. |
| Key outcome measure | COPD exacerbation rates. |
| Rationale for presentation | Conduct and follow-up, collecting data on benefits and harms. |
| Key discussion points | The study was the first in the world to have evaluated the effectiveness and safety of a prelicenced medicine compared with standard of care when used in every day clinical practice in patients with COPD. Patients were randomised at their routine GP respiratory review visit and constant monitored through the 12-month study via real time data collection from GP and hospitals. Data was also collected from participating community pharmacies and from Office For National Statistics mortality and Secondary User Service national datasets and an out of hours phone service. A safety alerting and reporting system was established based on serious adverse events being initially flagged in the EHR followed by review by the safety team and the principal investigator prior to submission to the sponsor as appropriate.<br>Lessons learnt were to ensure at the design phase that all the safety and endpoints of the study could be readily captured and the importance of early engagement with the regulators and National Institute of Health and Care Excellence. The study demonstrated that safety monitoring in this type of real-wold study was effective, could be done close to real time and was highly configurable. Widespread adoption could improve safety and such systems lend themselves to novel trial designs. |

COPD, chronic obstructive pulmonary disease; EHR, electronic healthcare record.

of participants and their GPs. Considerable efficiencies could have been possible if the HES data had been used as the main source of outcome information from the outset. Further benefits could be derived by accessing primary care data, as some cardiovascular events do not require hospitalisation and are consequently not captured in the HES data. The HES dataset is usually at least 3 months old. However, the 'rawer' form of HES, called Secondary Uses Service is being used in the RECOVERY trial to enable faster, although in extremis, assessment of health outcomes for patients with severe COVID-19 infection.

EHR can be used to assess the short-term and long-term impact of an intervention on health outcomes of individual patients at participating sites. This can be through a patient-level intervention or via a randomised cluster trial where the intervention is applied at a site level. One such example is the REDUCE cluster randomised trial (ISRCTN95232781) embedded with the CPRD GP practice research network, which used a digital intervention versus standard of care to attempt to reduce unnecessary use of antibiotics for respiratory infections in primary care. In addition to using the EHR to monitor prescribing rates, EHR data were used to monitor patient health outcomes in the intervention and standard of care GP practice arms of the trial.[11] Findings from ASCEND,[19 20] COPE[8] and DECIDE[11] support the advantages gained for data collection and follow-up by using EHR from the outset for patients recruitment.

The Office for National Statistics system has flagged people as participating in each of around 200 approved trials so that mortality data could be provided to the

trial in due course. All EHR systems need to include this ability to flag for participation in a specific trial. Without flagging, triallists and EHR providers need to go through the linkage process. This increases costs against limited budgets, is disproportionately effortful and exposes a repeated risk of linkage errors. Both Public Health England (PHE) National Cancer Registration and Analysis Service (NCRAS) and NHS Digital are piloting approaches to trial flagging.

### Collecting outcome data on benefits and harms

Access to timely data is key for any trial, and triallists must be able to report on safety and key outcome measures in a suitable timescale, including for the final analyses and for Data Monitoring Committee review of interim analyses. Trials must meet the stringent regulatory timelines set out in UK law, which includes reporting to the competent authority, appropriate suspected unexpected serious adverse reactions (SUSAR), within short timelines that starts when the sponsor of the trial receives the information from the clinical site.

The timely capture and reporting of certain safety outcome measures are critically important for the Clinical Trials of Investigational Medicinal Products trials run under the clinical trials regulations, where a subset of serious adverse events (SAEs) must be reported to the regulator within a short, prescribed timescale of 7 or 15 days. Researchers must quickly determine whether the SAE is a 'reaction' (Serious Adverse Reaction (SAR)) and, if so, whether this reaction is an 'unexpected' (SUSAR) according to clearly defined Reference Safety

Information. SAR and SUSAR are more likely in some treatment settings for example, cancer chemotherapy trials. 'Seriousness' definitions include, among others, 'hospitalisation' or 'prolongation of hospitalisation', which may or may not be captured across different EHR systems.

Frequency of updates to routine EHR data depends on where in the system the data are collected. Primary care records, such as those received by CPRD, are updated daily, but administrative-focused EHR datasets, such as HES, may provide only a snapshot of data from some months ago. Trials using only periodically updated, routinely collected EHR generally cannot rely on these data for reporting purposes in regulatory timescales, in particular for safety monitoring. It is a key responsibility of researchers to understand the strengths and limitations of any EHR data source, including its suitability to meet reporting requirements. A future involving interconnected real-time systems would allow investigators to reliably detect and report appropriate events.

The Salford Lung Studies were two industry-sponsored, late-phase randomised controlled trials (RCTs) that were the first studies in the world to have evaluated the effectiveness of a prelicenced medicine in a real-world setting. The study capitalised on access to primary care EHR, as well as community pharmacy data and electronic secondary care patient records embedded the regional hospitals. Altogether 7200 patients with COPD or asthma were monitoring in near real time for safety and outcomes using city-wide linked EHR.[21 22] The study team met with the regulators to agree the outcome measure data and safety alerting and reporting systems, which were predicated entirely on capture of routine data from near real-time primary and secondary care systems. As outlined in table 4, the studies successfully demonstrated that it is possible to run complex large-scale trials with configurable near real-time safety monitoring systems that meet regulatory standards.

It is important to check up-front, in efficacy and effectiveness trials that may be used for regulatory submission, whether routinely collected EHR would be accepted as source data by the relevant regulatory authorities. The Medicines and Healthcare products Regulatory Agency (MHRA) now considers using source data directly from each hospital's EHR standard following digitalisation of health records. GCP inspections performed by the MHRA in the UK are performed against the requirements of the UK Clinical Trials Regulations, irrespective of whether the source data are electronic or paper in format. The MHRA recognises that there is no standardised method of recording clinical source data, especially in international multicentre trials (P Walker, personal communication, 2020). Variations across sites include how many electronic systems make up the EHR, and how clinical data are reported through local laboratory electronic systems. It is therefore recommended each sponsor reviews the type of data required for their trial and how it would be recorded at site, including the various electronic systems

that capture the source data and whether they are fit for purpose.

The source data and its underpinning EHR system needs to follow the 'ALCOA+principles' to be attributable, legible, contemporaneous, original and accurate, as well as complete, consistent, enduring and available.[23] This ensures the data are reliable and that the electronic system is fit for its intended purpose. This should include appropriate access levels for investigator site trial teams, sponsor representatives and government regulators and is supported by the International Conference on Harmonisation E6 R2 Addendum. Careful work is required to ensure that systems that collate data from multiple healthcare systems (eg, HES or registry data) follow these principles. Review of the data on regulatory inspection by the MHRA includes assessing whether the trial data can be reconstructed and that the previous requirements are met. Reconstruction includes assessing whether the system has an associated audit trail verifying who completed assessments and when and the ability to reconstruct any changes to the source data. Source data verification performed during MHRA inspections is always done on a risk-based approach, with a focus on the critical data for the trial and compliance with the clinical trials regulations, recognising that EHR systems are often designed with standard care not clinical trials in mind. The same risk-based approach should be taken by sponsors auditing their own trial data.

### Analysis and interpretation

Many researchers desire collecting as much data as possible for their trial through routine EHR sources, thereby minimising the collection of data through trial-specific means. Reviewing metadata or deidentified samples of data, broadly representing the data that should be available for the sort of patients that could join a future trial, is recommended to determine which datasets include suitable variables to monitor health outcomes from EHR sources. In the case of cancer, PHE has developed Simulacrum to imitate some of the data held by NCRAS.[24] The Health Data Research Innovation Gateway has been initiated by HDR UK as an open repository of data with associated metadata to accelerate discovery and access to existing health data research resources.[25] Some argue that a common repository of data quality checks may be useful for researchers running data-enabled trials. Carrying out data utility assessments are a practical method of determining where EHR could substitute for trial-specific collection in future trials.[26 27]

Any single EHR dataset is not likely to collect all the necessary information, in a timely and consistent manner, that can be used to evaluate health outcomes across all trial arms. Table 5 sets out some of the lessons learnt during the Standard and New Antiepileptic Drugs II trial.[28 29] This trial for patients with epilepsy found that seizures, key to any epilepsy core outcome set, had not been well recorded in the accessible EHR sources. Initiatives such as the Core Outcome Measures in Effectiveness Trials

**Table 5** Key points from SANAD-II trial

| | |
|---|---|
| Trial name (registration) | SANAD-II[28 29] (ISRCTN30294119) |
| Sponsorship | Academic – Liverpool. |
| Clinical setting | Epilepsy. |
| Design | Randomised controlled trial. |
| Size | N=1510. |
| Key outcome measure | Time to 12 months without seizure. |
| Rationale for presentation | Analysis and interpretation. |
| Key discussion points | The SANAD-II trial. SANAD-II is a pragmatic, UK, multicentre, phase IV randomised controlled trial funded by the National Institute for Health Research Health Technology Assessment programme, assessing the clinical and cost-effectiveness of a number of antiepileptic drugs as first-line treatments for newly diagnosed epilepsy. The primary outcome measure related to a period without seizures. These data were recorded on case report forms (CRFs) by the treating clinical team at outpatient visits.<br>In a study of 98 patients, the team found the seizures were poorly recorded in EHR. Using only EHR data estimated that 74% of patients had spent 1 year seizure free, whereas using only CRF data estimated that only 46% of patients had spent 1 year seizure free. This is a clinically important difference. Researchers need to be able to trust in the completeness of EHR data to use them in reliable analyses or to understand how these data might supplement CRFs. This example cautions against naïve use of routinely collected EHR data. |

EHRs, electronic healthcare records; SANAD-II, Standard and New Antiepileptic Drugs II.

are seeking to develop sets of core outcome measures (COSs) that are suitable to support RCTs, as well as audit, research and routine care. Researchers developing COS should strive to make them as similar as possible to both the information required in routine practice and the key outcomes needed by regulatory reviewers.

Routinely collected datasets have contrasting and complementary strengths: hospital EHR datasets are rich in specific areas but lack the wider medical history for each patients; GP EHR datasets capture the broader, longitudinal medical history but not all the specialist depth; and registry datasets may have rich healthcare data but for only a narrow patient subgroup; and central administrative datasets collating information across healthcare systems may be wide but not deep. Therefore, it may be necessary to use multiple datasets to analyse trial outcomes, and these datasets may need to be linked. Well-established processes exist to link data to trials, although delays can emerge from governance and data access issues.

Data linkage can be used to validate data collected on an individual trial participant from trial-specific source or other routine datasets. The comparisons of multiple sources can support assessment of generalisability and external validity for future research. The use of alternative data sources may address any concerns about the representativeness of patients in a trial in comparison with those who, for whatever reason, did not participate (eg, trial not open in the area), including checking that the participants are broadly representative of the target population. An example of this is where primary data were used to validate the outcomes of the real-world Salford Lung Studies, in the broader UK population thereby ruling out any bias that may be introduced due to a regional Hawthorn effect.[30] For treatments

that are widely available outside of the research setting, researchers may assess external validity or representativeness of the patient populations if data can be requested for an equivalent non-randomised anonymised cohort of patients.[11]

For trials that combine multiple sources of data, trialists must prespecify what will be done in the instance of discrepancy between sources: which source would take primacy, who makes that decision, how checking will be done and whether there is any opportunity to correct one of the data sources, if required, in a manner consistent with data protection regulations about holding accurate data.

It is critically important to ensure that the correct EHR variables are both well understood and used in the right way to calculate the relevant outcome measures. Using multiple sources with differences in terminology and structure requires careful, prospective analytic approaches to derive meaningful information about the safety and efficacy of treatments.

## DISCUSSION

NHS datasets provide a record of population-level health data and afford rich opportunities for improving health through evidence-based medicine. There is a recognition that the UK can grasp these opportunities and address the challenges required to usher in a new generation of more efficient and cost-effective clinical trials. Many of these trials will be 'data-enabled trials', with routinely collected EHR driving their design and feasibility, their screening and recruitment and their management and analysis. Others will use EHR for more established practices such as long-term participant follow-up. The workshop demonstrated that EHR can readily be used to assess

trial feasibility and to facilitate patient recruitment from phase II to phase IV. However, attendees sensed that the short-term emphasis for conducting 'fully data-enabled trials' should be on licenced treatments in different populations or for new indications, where outcome measures are readily collectable from the EHR. Trials like the Salford Lung Studies and DECIDE serve as vanguards demonstrating the application of real-world evidence in fully data-enabled clinical trials.[17 21 22]

EHR data are collected in different formats and frequency depending on their purpose and provenance. There are multiple EHR collating-and-holding bodies within the UK, but no one body can provide access to all sources of EHR data. This situation results in a complicated web of access routes to datasets, differing in scope, method of collection, currency of data, quality controls, availability of metadata and connection to additional linked information. It is currently estimated that fewer than 5% of trials in the UK successfully access routinely collected datasets to support trial execution,[31] much lower than the proportion of trials planning to access such data at the grant application stage.[32] Therefore, more is clearly needed to increase confidence and capabilities in the use of routine health data across the UK clinical trials community. In the short term, route maps depicting these dataset details and access methods would be welcomed. In the long term, the trials community would be keen to see intraoperable EHR systems providing simple access to up-to-date datasets to facilitate trials: intraoperability and reusability emerged repeatedly as themes throughout the workshop. There are complex and varying coding systems across the various healthcare practice settings for which clarity is required. The UK Secretary of State has been aware of the need for better data standards in healthcare and has started to put NHS quality data into dashboards.[33] Many data items are held in free text so sophisticated natural language processing approaches will be required to extract coherent, targeted information, while upholding confidentiality standards. HDR UK's Health Data Research Innovation Gateway has progressed towards making more accessible the datasets routinely available datasets more accessible for a range of research uses,[34] although initially this resource is more likely to facilitate reuse of data from trials rather than supporting recruitment and conduct of new trials.

Gaining the necessary permissions to access EHR data causes challenges for many researchers, particularly when the interpretation of patient consent has changed over time with the strengthening information governance requirements. Consent from trial participants to access EHR data is taken in many trials only with the intention of accessing long-term outcome measures, so researchers may not submit the relevant applications until the linked data are needed, late in the trial. Unfortunately, expectations around consent might have changed by this time point, complicating the process. Researchers are therefore strongly recommended to access EHR data from early in the trial and for trial consent forms to prospectively cover broad and enduring consent for access to routine EHR as a matter of course in order to facilitate research.[35] By embedding the EHR from the design stage, with appropriate consent, data-enabled trials avoid this issue. A more streamlined approach to governance with greater consistency and clarity in terminology and requirements for enduring patient consent for research may facilitate future data access for all trials. Where additional data are required and collected for a clinical trial, there may be potential to augment the patient's EHR with this information, such as results of genetic or genomic tests or patient-reported outcome measures.

Clinical trials are expensive and slow to plan and complete, and it is necessary to reduce research waste through all possible methods.[36–39] Trial design takes at least 6 months yet is often insufficiently resourced, relying on 'own account' work taken at the risk of the sponsor (usually a higher education institute) prior to grant award. A substantial proportion of trials are delayed or do not reach their recruitment target,[9 40] and a considerable number require at least one protocol amendment.[41] EHR offer opportunities to better plan clinical trials by characterising the populations and their treatment or diagnostic pathways as well as assessing the trial feasibility based around specific geographies.[8] The proportion of patients approached and recruited to trials also varies according to different demographics such as ethnicity or age.[42 43] Using EHR data to identify specific patient pools will enable trials in underserved populations or where disease burden is the greatest, thereby democratising research and increasing the potential recruitment efficiencies.[44] Increasing the representativeness of trial participants also enhances external validity of the findings, especially as guidance such as the 'Innovations in Clinical Trial Design and Delivery for the Under-served' (INCLUDE) framework is now available on the issues to consider to be more inclusive on attributes such as ethnicity.[45] UK-wide representative population coverage, through CPRD, has demonstrated the proof of principle of successfully recruiting a demographically representative patient population into the pragmatic DECIDE in type 2 diabetes trial, through centralised primary care patient data eligibility searches (J Valentine, personal communication, 2020). Virtually all of the UK population is registered with a GP practice, and GPs are gatekeepers to the NHS, often managing chronic conditions that have been diagnosed in secondary care. Barriers to GPs' participation in primary healthcare research are well recognised.[46 47] We anticipate that new, data-enabled methods will increase awareness of research participation for both patients and GPs. The role of GPs acting as potential 'gatekeepers' to research will be an important issue, recognising that ethics committees or regulators may require the approval of a clinician who has responsibility for the patient in some studies. With their rich, longitudinal medical history, primary care datasets represent an excellent source of data to locate suitable patients for trials based in both primary and secondary care settings.

**Table 6** Key points from ASCEND trial

| | |
|---|---|
| Trial name (registration) | ASCEND: A Study of Cardiovascular Events iN Diabetes[19 20] (NCT00135226) |
| Sponsorship | Academic – Oxford. |
| Clinical setting | Diabetes mellitus, UK only: national, mail-based methods – no study sites. |
| Design | 2×2 factorial design randomised placebo-controlled trial (phase 4). |
| Size | n=15 000. |
| Key outcome measure | Cardiovascular events. |
| Rationale for presentation | Recruitment, conduct and follow-up. |
| Key discussion points | ASCEND used highly streamlined mail-based methods to identify, recruit and follow-up (for an average of 7.5 years) 15 500 UK patients, making it one of the longest duration and largest ever trials in diabetes.<br>Participants completed 6-monthly follow-up questionnaires, initially using paper, but later moving to an online system for those who wished.<br>The trialists gained access to HES data during the closing stages of the study, allowing additional follow-up information to be gained for those who had not returned a recent questionnaire.<br>Subsequent analyses of the HES data compared with questionnaire data showed good concordance, suggesting that, in the UK, HES data alone could be used as a highly efficient means of follow-up for cardiovascular trials in the future. |

HES, Hospital Episode Statistics.

A strong desire is a single resource that brings together all primary care data within and across all four UK nations.

Validation of the choice of EHR codes used in patient cohort searches enables greater specificity of patient searches, leading to higher patient conversion rates into and less time lost by patients and trial teams at the trial screening stage (J Wilding, personal communication, 2020). Clinical validation of potentially suitable patients initially flagged by EHR searches further refines the patient search, meaning that the patients approached to take part are more likely to pass eligibility screening, which improves the efficiency of patient recruitment. Assessment of the variables contained within the EHR at the trial design stage enables a better understanding of which variables can be obtained from routinely collected data, removing duplication of collection. Moving from only trial-specific data collection to more widespread use of EHR within trials should improve the cost-effectiveness of trial funding. Double-blinding and placebo controls are more difficult aspects of conduct to operationalise in routine EHR settings but may be offset by robust data, careful protocols and objective outcome measures. Each trial's risk assessment should be clear on any risks associated with the use of routine EHR data. Formal regulatory advice meetings may provide clarification of the proposed approaches, including monitoring and data verification. SAEs require construction by the investigators of an accompanying narrative that may require detail beyond that collected by the research team access routinely collected EHR source but which should be available to the responsible investigator.

Methodological engagement, such as from MRC-NIHR Trials Methodology Research Partnership,[48] will help to assess the settings for which using routinely collected data are, or are not, more suitable than trial-specific data collection. A Study Within A Trial (SWAT) approach[49–51] can efficiently assess particular elements of EHR trials, such as from the WOSCOPS trial.[52 53] Collation of findings from retrospective comparisons will elucidate issues for prospective clarification and highlight challenges to address around better data collection.[26]

The impact of 2020's SARS-CoV-2 pandemic was keenly felt by clinical trials with many hindered by a necessary pause, postponement or termination of recruitment and/ or treatment. Routine hospital appointments became virtual and many research assessments had to follow. While trials teams could pivot to collect data online directly from participants, postponed or cancelled scans and tests will cause analysis and interpretation issues through

**Box 3    Recommendations for data-enabled trials**

► Wider adoption of data-enabled methods for clinical trials.
► Build in methodology questions into the trial protocol, for example, conversion rates for identification to recruitment; refining the process for identifying potential patients; understanding event rates for the target population.
► Early engagement of regulators with regards to monitoring and data verification.
► Patients and public involvement to robustly inform data-enabled trials.
► Increased interoperability of electronic health record (EHR) systems.
► Clear map for access to EHR: what is captured where, what do the data mean and how can they be used.
► Consistency between core outcome sets used for research and routine practice.
► Wider sharing and reporting of EHR code lists to support recruitment and outcome measures.
► Sharing of detailed examples of success stories.
► Capacity-building and capability-building for future data-enabled trials.

the trial literature for many years, which will need to be reported systematically using the extension to Consolidated Standard for Reporting Trials standards currently in development. National strategic prioritisation of trials opened doors to routinely collected data with uncommon rapidity. This may allow the development of new access routes to data for trials and more widespread adoption by the research community of EHR-based methodology within trial delivery.

The UK has led by example in patients and public involvement and engagement (PPIE) for trials,[54 55] and key funding bodies support clear PPIE. Patient ambassadors may be particularly helpful in informing the development of data-enabled trials. Locating participants via their EHR will enable participants to be informed of the trial's results which, surprisingly, still does not always happen, although there is limited information available on best practice to inform participants; the Show Respect study is addressing this.[56] The StatinWISE study (NCT02781064), a non-randomised cohort N-of-1 study of stopping statins in patients who have had heart attacks, presented each patient a booklet of their own results before the study reported overall findings.[57] There is a clear need to engage the public on clinical trials, sensitising communities to research and with a specific focus on the capacity of EHR to provide better outcomes.

This workshop showcased key successes and high-profile case studies (tables 2–6) that have answered questions where EHR data could map well to the trial questions, supplemented by some trial-specific collection. Overinterpretation of the clinical trials regulations can be risk averse and suffocate innovation. Simplicity in implementation need not be interpreted as basic in ambition: straightforward protocols can still address a number of key clinical questions simultaneously.[1 58] Knowledge-sharing and capability-building exercises now should facilitate future data-enabled trials. Triallists are getting better at writing up their findings and, increasingly, their designs, but further effort is required in writing up their practical implementation experiences. Best practice and case studies, reflecting on successes and challenges, should be shared, including in the threaded publications from NIHR Health Technology Assessment monographs[59] or a new, dedicated national forum.

Five years previously, NIHR launched a call for trials using routinely collected data that elicited a modest response form research community. Stimulating research using routinely collected clinical data has remained a sustained interest of NIHR, and launching out from this workshop, NIHR announced a new call for data-enabled trials.[4] NIHR committees reviewed 18 applications and the funded trials were confirmed in 2020 (table 1).

Routinely collected EHR offer great potential for faster, more efficient and less expensive trials reaching a broader catchment of the UK population. Wider adoption of EHR-based approaches and experiences from these studies, such as those funded from the NIHR data-enabled trial call, will provide the evidence to assess where EHR enablement brings benefits to trial execution and follow-up. Key recommendations from the workshop are presented in box 3, many of which are now being addressed. Using EHR data for trials can be done and could be done at a greater scale than is the current practice. The desire to access and efficiently use these data is widely discussed, and this workshop demonstrated that this aspiration is possible. The opportunities to learn and harness the power of information and technology to improve healthcare outcomes for everyone has never been better.

**Author affiliations**
[1]MRC Clinical Trials Unit at UCL, Institute of Clinical Trials and Methodology, University College London, London, UK
[2]Medicines and Healthcare products Regulatory Agency (MHRA), London, UK
[3]MRC Clinical Trial Service Unit and Epidemiological Studies Unit, Nuffield Department of Population Health, University of Oxford, Oxford, UK
[4]NHS Digital, Leeds, UK
[5]Nuffield Department of Primary Care Health Sciences, University of Oxford, Oxford, UK
[6]Trials and Studies Coordinating Centre, National Institute for Health Research Evaluation, Southampton, UK
[7]North West E-Health (NWEH) Ltd, Manchester, UK
[8]King's College London, London, UK
[9]NIHR Biomedical Research Centre at Guy's and St Thomas' Hospitals London, London, UK
[10]Intensive Care National Audit & Research Centre (ICNARC), London, UK
[11]York Trials Unit, Department of Health Sciences, The University of York, York, UK
[12]Lancaster University, Lancaster, UK
[13]Health Research Authority, London, UK
[14]Edinburgh Clinical Trials Unit, University of Edinburgh, Edinburgh, UK
[15]Department of Respiratory Epidemiology, Occupational Medicine and Public Health, Imperial College London, London, UK
[16]NIHR Health Services & Delivery Programme, Southampton, UK
[17]NIHR Clinical Research Network, University of Leeds, Leeds, UK
[18]Department of Epidemiology and Population Health, London School of Hygiene & Tropical Medicine, London, UK
[19]Division of Population & Behavioural Science, University of St. Andrews, St Andrews, UK
[20]Department of Family & Community Medicine, University of Toronto, Toronto, Ontario, Canada
[21]Department of Cardiovasular and Metabolic Medicine, Institute of Life Course and Medical Sciences, University of Liverpool, Liverpool, UK
[22]Department of Health Data Science, University of Liverpool, Liverpool, UK
[23]Nuffield Department of Population Health, University of Oxford, Oxford, UK
[24]NIHR Oxford Biomedical Research Centre, Oxford University Hospitals NHS Foundation Trust, Oxford, UK
[25]Health Data Research UK, University of Oxford, Oxford, UK
[26]Health Data Research UK, London, UK
[27]University of Nottingham, Nottingham, UK
[28]Director of the NIHR Health Technology Assessment Programme (2015-2020), Southampton, UK
[29]Clinical Practice Research Datalink, Medicines and Healthcare Products Regulatory Agency, London, UK

**Collaborators** The Data Enabled Trials Group Workshop Group members (name, ORCID (if known), affiliation) are: Deborah Ashby (0000-0003-3146-7466) (Imperial College London); Yolanda Barbachano (Medicines and Healthcare products Regulatory Agency (MHRA)); Susan Beatty (Clinical Practice Research Datalink); Marion Bennie (HDR Scotland); Helen Bodmer (Medical Research Council); Louise Bowman (0000-0003-1125-8616) (Clinical Trial Service Unit and Epidemiological Studies Unit, Nuffield Department of Population Health, University of Oxford, Oxford); Emer Brady (0000-0002-4715-9145) (Leicester General Hospital); Paul Brocklehurst (North Wales Organisation for Randomised Trials); Brooke Jackson (Clinical Practice Research Datalink); Marion Campbell (0000-0001-5386-4097)

(HDR Scotland); Andrea Cipriani (0000-0001-5179-8321) (University of Oxford); Jane Daniels (0000-0003-3324-6771) (University of Nottingham); Tom Denwood (0000-0002-7337-2425) (NHS Digital); Michael Donnelly (Queen's University Belfast); Tobias Dreischulte (University of Dundee); Mark Edwards (0000-0002-8616-3753) (University Hospital Southampton NHSFT); Andrew Farmer (0000-0002-6170-4402) (Nuffield Department of Primary Health Care Sciences, University of Oxford.); Nick Freemantle (0000-0001-5807-5740) (Comprehensive Clinical Trials Unit, UCL); Steph Garfield-Birkbeck (0000-0002-3181-2840) (National Institute for Health Research Evaluation, Trials and Studies Coordinating Centre); Martin Gibson (0000-0002-1331-1524) (North West E-Health (NWEH) Ltd); Steve Goodacre (0000-0003-0803-8444) (National Institute of Health Research Health Technology Assessment CET); Doug Gould (0000-0003-4148-3312) (Intensive Care National Audit & Research Centre (ICNARC)); Laura Gray (0000-0002-9284-9321) (University of Leicester); Xavier Griffin (0000-0003-2976-7523) (HDR UK Oxford); Martin Gulliford (0000-0003-1898-9075) (Kings College London, London); David Harrison (0000-0002-9002-9098) (Intensive Care National Audit & Research Centre (ICNARC), London, UK); Catherine Hewitt (0000-0002-0415-3536) (York Trials Unit, Department of Health Sciences, University of York); Thomas Hiemstra (0000-0002-2115-8689) (University of Cambridge); Anna Higgins (University of Nottingham); Julia Hippisley-Cox (0000-0002-2479-7283) (University of Oxford); Michael King (0000-0003-4715-7171) (PRIMENT Clinical Trials Unit, UCL); Sasha Korniak (Intensive Care National Audit & Research Centre (ICNARC)); Martin Landray (0000-0001-6646-827X) (Nuffield Department of Population Health, University of Oxford, Oxford, UK; Health Data Research UK, University of Oxford, Oxford, UK; National Institute of Health Research Oxford Biomedical Research Centre, Oxford University Hospitals NHS Foundation Trust, Oxford, UK; Martyn Lewis (0000-0002-3667-132X) (Keele Clinical Trials Unit); Fiona Lobban (0000-0001-6594-4350) (Lancaster University); Jennifer Logue (0000-0001-9549-2738) (Lancaster University); Guillermo Lopez Campos (0000-0003-3011-0940) (Queen's University Belfast); Stephanie MacNeill (University of Bristol); Marion Mafham (0000-0003-0562-3963) (HDR UK Oxford); Colin McCowan (0000-0002-9466-833X) (HDR UK Scotland); Kathleen Meeley ((Previously) Medicines and Healthcare products Regulatory Agency (MHRA)); Agnieszka Michael (0000-0002-7262-6227) (Surrey Trials Unit); Nick Mills (BHF Centre for Cardiovascular Sciences); Andrew Morris (0000-0002-1766-0473) (Health Data Research UK); Will Navaie (Health Research Authority); Irwin Nazareth (0000-0003-2146-9628) (PRIMENT Clinical Trials Unit, UCL); Chris Newby (Queen Mary University of London); John Norrie (0000-0001-9823-9252) (Edinburgh Clinical Trials Unit, University of Edinburgh); Jacqui Nuttall (Southampton Clinical Trials Unit); Martin O'Kane (Medicines and Healthcare products Regulatory Agency (MHRA)); Jennifer Quint (0000-0003-0149-4869) (NHLI, Imperial College London, UK); Dheeraj Rai (0000-0002-7239-3523) (University of Bristol); Ian Roberts (0000-0003-1596-6054) (London School of Hygiene & Tropical Medicine Clinical Trials Unit); Mike Robling (0000-0002-1004-036X) (University of Cardiff); Jo Rycroft-Malone (0000 0003 3858 5625) (Lancaster University & National Institute of Health Research Health Services & Delivery Programme); Jonathan Sheffield (Formerly CEO National Institute of Health Research Clinical Research Network; University of Leeds); Aziz Sheikh (0000-0001-7022-3056) (HDR UK (Scotland)); Murali Shyamsundar (0000-0003-3797-8080) (Queen's University Belfast); Liam Smeeth (0000-0002-9168-6022) (London School of Hygiene and Tropical Medicine); Claire Snowdon (ICR Clinical Trials & Statistics Unit (ICR-CTSU)); Jamie Soames (0000-0001-8948-3480) (National Institute of Health Research); Derek Stewart (PPI); Adam Streeter (University of Plymouth); Frank Sullivan (0000-0002-6623-4964) (Division of Population & Behavioural Science, University of St. Andrews; Department of Family & Community Medicine, University of Toronto); Matthew Sydes (0000-0002-9323-1371) (MRC Clinical Trials Unit at UCL, Institute of Clinical Trials and Methodology, UCL, London, UK & HDR UK); Doreen Tembo (0000-0002-8615-3563) (National Institute of Health Research Evaluation, Trials and Studies Coordinating Centre (NETSCC)); Juliet Tizzard (Health Research Authority); Catrin Tudur Smith (0000-0003-3051-1445) (University of Liverpool); Janet Valentine (Clinical Practice Research Datalink); Paula Walker (Medicines and Healthcare products Regulatory Agency (MHRA)); Rhoswyn Walker (0000-0001-5605-2879) (Health Data Research UK); Alan Watkins (University of Swansea); Will Whiteley (0000-0003-2839-8404) (Department of Cardiovascular and Metabolic Medicine; Institute of Life Course and Medical Sciences; University of Liverpool); John Wilding (0000-0003-2839-8404) (Department of Cardiovascular and Metabolic Medicine, Institute of Life Course and Medical Sciences, University of Liverpool); Hywel Williams (0000-0002-5646-3093) (Professor of Dermato-Epidemiology, University of Nottingham); Kate Williams (University of Cambridge); Elizabeth Williamson (0000-0001-6905-876X) (London School of Hygiene & Tropical Medicine); AND Paula Williamson (0000-0001-9802-6636) (University of Liverpool).

**Contributors** Organised workshop: ML, AM, JV, RRW and HCW. Attended workshop: YB, LB, TD, AF, SG-B, MG, MCG, DAH, CH, ML, JL, AM, WN, JN, MO, JKQ, JR-M, JS, LS, FS, MRS, JT, JV, RRW, JW, HCW and PRW. Contributed to workshop discussion: YB, LB, TD, AF, SG-B, MG, MCG, DAH, CH, ML, JL, AM, WN, JN, MO, JKQ, JR-M, JS, LS, FS, MRS, JV, RRW, JW, HCW and PRW. Gave plenary presentation: LB, TD, MG, MCG, ML, MO, JKQ, LS, JV, JW and PRW. Chaired session: AF, ML and AM. Chaired discussion group: DAH, CH, JL, JN, FS and PRW. Member of discussion panel: YB, WN, JR-M, JS and JT. Scribed workshop: SG-B and MRS. Scribed discussion group: DAH, CH, JL, JN and FS. Wrote key sections of manuscript: MRS and JV. Led writing of manuscript: ML, MO, MRS, JV, PW, RRW and PRW. Contributed to and agreed manuscript: YB, LB, TD, AF, SG-B, MG, MCG, DAH, CH, ML, JL, AM, WN, JN, MO, JKQ, JR-M, JS, LS, FS, MRS, JV, PW, RRW, JW, HCW and PRW.

**Funding** The workshop was cofunded by National Institute of Health Research Health Technology Assessment Programme, Clinical Practice Research Datalink and Health Data Research UK; MRC grant MC_UU_12023/24 funded the salary of Matthew Sydes.

**Competing interests** JW reports grants, personal fees and other from AstraZeneca, other from Astellas, other from Boehringer Ingelheim, other from Lilly, other from Napp, grants, personal fees and other from Novo Nordisk, other from Sanofi, grants, personal fees and other from Takeda, other from Rhythm Pharma, outside the submitted work.

**Patient consent for publication** Not required.

**Provenance and peer review** Not commissioned; externally peer reviewed.

**ORCID iDs**
Matthew R Sydes http://orcid.org/0000-0002-9323-1371
Louise Bowman http://orcid.org/0000-0003-1125-8616
Tom Denwood http://orcid.org/0000-0002-7337-2425
Andrew Farmer http://orcid.org/0000-0002-6170-4402
Steph Garfield-Birkbeck http://orcid.org/0000-0002-3181-2840
Martin Gibson http://orcid.org/0000-0002-1331-1524
Martin C Gulliford http://orcid.org/0000-0003-1898-9075
David A Harrison http://orcid.org/0000-0002-9002-9098
Catherine Hewitt http://orcid.org/0000-0002-0415-3536
Jennifer Logue http://orcid.org/0000-0001-9549-2738
John Norrie http://orcid.org/0000-0001-9823-9252
Jennifer K Quint http://orcid.org/0000-0003-0149-4869
Jo Rycroft-Malone http://orcid.org/0000-0003-3858-5625
Liam Smeeth http://orcid.org/0000-0002-9168-6022
Frank Sullivan http://orcid.org/0000-0002-6623-4964
John Wilding http://orcid.org/0000-0003-2839-8404
Paula R Williamson http://orcid.org/0000-0001-9802-6636
Martin Landray http://orcid.org/0000-0001-6646-827X
Andrew Morris http://orcid.org/0000-0002-1766-0473
Rhoswyn R Walker http://orcid.org/0000-0001-5605-2879
Hywel C Williams http://orcid.org/0000-0002-5646-3093

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
