## [Reviewer comments · BMJ Open]

ARTICLE DETAILS

TITLE (PROVISIONAL)	Realising the full potential of data-enabled trials in the UK: a call for action
AUTHORS	Sydes, Matthew; Barbachano, Yolanda; Bowman, Louise; Denwood, Tom; Farmer, Andrew; Garfield-Birkbeck, Steph; Gibson, Martin; Gulliford, Martin; Harrison, David; Hewitt, Catherine; Logue, Jennifer; Navaie, Will; Norrie, John; O'Kane, Martin; Quint, Jennifer; Rycroft-Malone, Jo; Sheffield, Jonathan; Smeeth, Liam; Sullivan, Frank; Tizzard, Juliet; Walker, Paula; Wilding, John; Williamson, Paula; Landray, Martin; Morris, Andrew; Walker, Rhoswyn; Williams, HC; Valentine, Janet; Data Enabled Trials Workshop Group, And the

VERSION 1 – REVIEW

REVIEWER	Simon, Gregory Kaiser Permanente Washington Health Research Institute
REVIEW RETURNED	11-Jan-2021

GENERAL COMMENTS	The topic is of interest to a broad research and policy audience. The presentation is generally clear, including recommendations useful to a range of stakeholders. I have some specific suggestions regarding the presentation. 1) In the section regarding recruitment (authors' page 11) the discussion of volunteer registries does not see relevant to or consistent with the authors' main point. Aren't registries of volunteers simply a more efficient tool to support the old regime rather than part of the new one the authors propose? 2) The mention of GPs vetting or approving research invitations does raise questions about the appropriateness of this gatekeeping function. Should a GP always have power to speak for their patients access to research participation? 3) The section regarding Conduct and Follow-Up does not seem (at least to me) clearly distinct from the preceding section regarding recruitment and the subsequent section regarding outcome assessment. Does this section address something specific and unique? Or does it just include examples of use of records data for the purposes discussed before and after? 4) The section regarding analysis and interpretation could be clearer. Perhaps the authors could more clearly distinguish questions regarding timeliness, harmonizing/combining data from multiple sources, and assessing representativeness or generalizability.
--

REVIEWER	Wood, William University of North Carolina at Chapel Hill School of Medicine
REVIEW RETURNED	15-Feb-2021

GENERAL COMMENTS	This is an adequately revised manuscript and makes an important contribution to the scientific literature.
--

VERSION 1 – AUTHOR RESPONSE

Reviewer: 1	
The topic is of interest to a broad research and policy audience. The presentation is generally clear, including recommendations useful to a range of stakeholders. I have some specific suggestions regarding the presentation.	We thank the reviewer for this thoughtful and constructive comments.
1) In the section regarding recruitment (authors' page 11) the discussion of volunteer registries does not see relevant to or consistent with the authors' main point. Aren't registries of volunteers simply a more efficient tool to support the old regime rather than part of the new one the authors propose?	We agree that we could have described the added value of registries more clearly. Registries may have additional data on patient phenotype or genotype beyond the information in clinical or administrative datasets. We have added a clarification so the sentence has changed – From: “Alternative methods for recruiting patients into prospective trials include establishing a cohort of patients or volunteers who are willing to take part in trials, such as the NHS Scotland SHARE scheme¹³ and COVID-19 vaccine trials portal,¹⁴ and pre-consenting trial participants to be re-approached about other studies, such as those following on from the INTERVAL trial (NCT01610635) of blood donation timing.¹⁵”

COMMENTS	RESPONSE
	To: “Alternative methods for recruiting patients into prospective trials include establishing a cohort of patients or volunteers who are willing to take part in trials, such as the NHS Scotland SHARE scheme¹³ and COVID-19 vaccine trials portal,¹⁴ – volunteer datasets may contain phenotype or genotype data not available in clinical or administrative datasets -- and pre- consenting trial participants to be re-approached about other studies, such as those following on from the INTERVAL trial (NCT01610635) of blood donation timing.¹⁵“
2) The mention of GPs vetting or approving research invitations does raise questions about the appropriateness of this gatekeeping function. Should a GP always have power to speak for their patients access to research participation?	We appreciate the reviewer’s challenge. We suspect that “always” is a difficult term, but note that GPs are the data controllers for access to their EHRs. In some studies, ethics committees or regulators may require the approval of a clinician who has responsibility for the patient to minimise potential harm. We have added a new sentence into the Discussion with two new references -- Added: “Barriers to GPs’ participation in primary health care research are well recognised. We anticipate that new data -enabled methods will increase awareness of research participation for both patients and GPs. The role of GPs acting as potential “gatekeepers” to research will be an important issue,

	recognising that ethics committees or regulators may require the approval of a clinician who has responsibility for the patient in some studies.”
3) The section regarding Conduct and Follow-Up does not seem (at least to me) clearly distinct from the preceding section regarding recruitment and the subsequent section regarding outcome assessment. Does this section address something specific and unique? Or does it just include examples of use of records data for the purposes discussed before and after?	We understand the point being made by the reviewer, and note that the other reviewer did not make this comments. The structure of the paper reflects the structure of the workshop. We had anticipated, and still maintain, that most aspects of conduct and follow-up are different to those of recruitment (identification of patients is a separate activity to following up participants for outcome measures). However, there is inevitably some overlap in the issues that are raised. In developing the manuscript, the authors discussed the structure at considerable length, and decided that it was appropriate to use the current structure that follows the sequence of a typical trial as the alternatives have more considerable limitations, including section length. We have not made any updates to the text.
4) The section regarding analysis and interpretation could be clearer. Perhaps the authors could more clearly distinguish questions regarding timeliness, harmonizing/combining data from multiple sources, and assessing representativeness or generalizability.	We agree there is scope to make this clearer still and have made a series of changes throughout the section to address this, including further clarification of the features of different sources of routine data with regard to comprehensiveness and timeliness and how these data can be combined and used for validation of participant representativeness. Changes in this section include these examples --

From:

“Data linkage can be used to validate data collected on an individual trial participant from trial-specific source or other routine datasets. The comparisons of multiple sources can support assessment of generalisability and external validity for future research. The use of alternative data sources may address any concerns about the representativeness of patients in a trial in comparison to those who, for whatever reason, did not participate (e.g. trial not open in the area), including checking that the participants are broadly representative of the target population.”

To:

“Data linkage can be used to validate data collected on an individual trial participant from trial-specific source or other routine datasets. The comparisons of multiple sources can support assessment of generalisability and external validity for future research. The use of alternative data sources may address any concerns about the representativeness of patients

in a trial in comparison to those who, for whatever reason, did

not participate (e.g. trial not open in the area), including

checking that the participants are broadly representative of the target population.”

And from:

“Using multiple sources with their current differences in terminology and structure could increase the likelihood of error.”

To:

“Using multiple sources with differences in terminology and structure requires careful, prospective analytic approaches to derive meaningful information about the safety and efficacy of treatments.”

For simplicity, we have not listed the other changes here but there should be clear in the tracked-changes document.

Reviewer: 2	
This is an adequately revised manuscript and	We appreciate this comment.
makes an important contribution to the	
scientific literature.	

VERSION 2 – REVIEW

REVIEWER	Simon, Gregory Kaiser Permanente Washington Health Research Institute
REVIEW RETURNED	27-Apr-2021
GENERAL COMMENTS	All of my questions and suggestions have been adequately addressed.